# DEMYSTIFYING RESNET

**Sihan Li**
Department of Electronic Engineering
Tsinghua University
Beijing 100084, China
`lisihan13@mails.tsinghua.edu.cn`

**Jiantao Jiao, Yanjun Han, Tsachy Weissman**
Department of Electrical Engineering
Stanford University
Stanford, CA 94305, USA
`{jiantao,yjhan,tsachy}@stanford.edu`

## ABSTRACT

We provide a theoretical explanation for the great performance of ResNet via the study of deep linear networks and some nonlinear variants. We show that with or without nonlinearities, by adding shortcuts that have depth two, the condition number of the Hessian of the loss function at the zero initial point is depth-invariant, which makes training very deep models no more difficult than shallow ones. Shortcuts of higher depth result in an extremely flat (high-order) stationary point initially, from which the optimization algorithm is hard to escape. The 1-shortcut, however, is essentially equivalent to no shortcuts. Extensive experiments are provided accompanying our theoretical results. We show that initializing the network to small weights with 2-shortcuts achieves significantly better results than random Gaussian (Xavier) initialization, orthogonal initialization, and shortcuts of deeper depth, from various perspectives ranging from final loss, learning dynamics and stability, to the behavior of the Hessian along the learning process.

## 1 INTRODUCTION

Residual network (ResNet) was first proposed in He et al. (2015a) and extended in He et al. (2016). It followed a principled approach to add shortcut connections every two layers to a VGG-style network (Simonyan & Zisserman, 2014). The new network becomes easier to train, and achieves both lower training and test errors. Using the new structure, He et al. (2015a) managed to train a network with 1001 layers, which was virtually impossible before. Unlike Highway Network (Srivastava et al., 2015a;b) which not only has shortcut paths but also borrows the idea of gates from LSTM (Sainath et al., 2013), ResNet does not have gates. Later He et al. (2016) found that by keeping a clean shortcut path, residual networks will perform even better.

Many attempts have been made to improve ResNet to a further extent. "ResNet in ResNet" (Targ et al., 2016) adds more convolution layers and data paths to each layer, making it capable of representing several types of residual units. "ResNets of ResNets" (Zhang et al., 2016) construct multi-level shortcut connections, which means there exist shortcuts that skip multiple residual units. Wide Residual Networks (Zagoruyko & Komodakis, 2016) makes the residual network shorter but wider, and achieves state of the art results on several datasets while using a shallower network. Moreover, some existing models are also reported to be improved by shortcut connections, including Inception-v4 (Szegedy et al., 2016), in which shortcut connections make the deep network easier to train.

Why are residual networks so easy to train? He et al. (2015a) suggests that layers in residual networks are learning residual mappings, making them easier to represent identity mappings, which prevents the networks from degradation when the depths of the networks increase. However, Veit et al. (2016) claims that ResNets are actually ensembles of shallow networks, which means they do not solve the problem of training deep networks completely.

We propose a theoretical explanation for the great performance of ResNet. We concur with He et al. (2015a) that the key contribution of ResNet should be some special structure of the loss function that makes training very deep models no more difficult than shallow ones. Analysis, however, seems non-trivial. Quoting He et al. (2015a):

> "But if $\mathcal{F}$ has only a single layer, Eqn.(1) is similar to a linear layer: $y = W_1 x + x$, for which we have not observed advantages. "

> *"Deeper non-bottleneck ResNets (e.g., Fig. 5 left) also gain accuracy from increased depth (as shown on CIFAR-10), but are not as economical as the bottleneck ResNets. So the usage of bottleneck designs is mainly due to practical considerations. We further note that the degradation problem of plain nets is also witnessed for the bottleneck designs. "*

Their empirical observations are inspiring. First, the 1-shortcuts mentioned in the first paragraph do not work. Second, noting that the *non-bottleneck* ResNets have 2-shortcuts, but the bottleneck ResNets use 3-shortcuts, one sees that shortcuts with depth three also do not work. Hence, a reasonable theoretical explanation must be able to distinguish the 2-shortcut from shortcuts of other depths, and clearly demonstrate why the 2-shortcuts are special and are able to ease the optimization process so significantly for deep models, while shortcuts of other depths may not do the job.

Aiming at explaining the performance of 2-shortcuts, we need to eliminate other variables that may contribute to the success of ResNet. Indeed, one may argue that the deep structure of ResNet may give it better representation power (lower approximation error), which contributes to lower training errors. To eliminate this effect, we focus on deep linear networks, where deeper models do not have better approximation properties. The special role of 2-shortcuts naturally arises in the study.

FIX

## 2 MAIN RESULTS

Our work reveals that *non-degenerate depth-invariant initial condition numbers, a unique property of residual networks with 2-shortcuts, contributed to the success of ResNet*. In fact, in a linear network that will be defined rigorously later, the condition number of Hessian of the Frobenius loss function at the *zero* initial point is

$$\text{cond}(H) = \sqrt{\text{cond}((\Sigma^{XX} - \Sigma^{YX})^T(\Sigma^{XX} - \Sigma^{YX}))}, \tag{1}$$

which is independent of the number of layers. Here $\Sigma^{XX}$ and $\Sigma^{YX}$ denote the input-input and the output-input correlation matrices, defined in Section 3.3. The condition number of a possibly non-PSD matrix is defined as:

**Definition 1.** The condition number of a matrix $A$ is defined as

$$\text{cond}(A) = \frac{\sigma_{\max}(A)}{\sigma_{\min}(A)}, \tag{2}$$

where $\sigma_{\max}(A)$ and $\sigma_{\min}(A)$ are the maximum and minimum of singular values of $A$. In particular, if $A$ is normal, i.e. $A^T A = A A^T$, the definition can be simplified to [1]

$$\text{cond}(A) = \frac{|\lambda(A)|_{\max}}{|\lambda(A)|_{\min}}, \tag{3}$$

where $|\lambda(A)|_{\max}$ and $|\lambda(A)|_{\min}$ are the maximum and minimum of the absolute values of eigenvalues of $A$.

Moreover, the *zero* initial point for ResNet with 2-shortcuts is in fact a so-called *strict* saddle point (Ge et al., 2015), which are proved to be easy to escape from.

Why shortcuts of other depths do not work? We show that the Hessian at the *zero* initial point for the 1-shortcut ResNet has condition number growing unboundedly for deep nets. As is well known in convex optimization theory, large condition numbers can have enormous adversarial impact on the convergence of first order methods (Nemirovski, 2005). Hence, it is quite clear that starting training at a point with a huge condition number would make the algorithm very difficult to escape from the initial point, making 1-shortcut ResNet no better than conventional approaches.

---

[1] The equivalence of Equation (2) and Equation (3) can be proved easily using the eigenvalue decomposition of $A$. Note that as Hessians are symmetric (if all the second derivatives are continuous), we will use Equation (3) for their condition numbers. As the $|\lambda|_{\min}$ of Hessian is usually very unstable, we calculated $\frac{|\lambda|_{\max}}{|\lambda|_{(0.1)}}$ to represent condition numbers instead, where $|\lambda|_{(0.1)}$ is the 10[th] percentile of the absolute values of eigenvalues. Note that the Hessian at zero initial point for 2-shortcut ResNet also has a nice structure of spectrum: see Theorem 1 for details.

For shortcuts with depth deeper than two, the Hessian at the *zero* initial point is a *zero* matrix, making it a higher-order stationary point. Intuitively, the higher order the stationary point is, the harder it is to escape from it. Indeed, it is supported both in theory (Anandkumar & Ge, 2016) and by our experiments.

One may still ask: why are we interested in the Hessian at the *zero* initial point? It is because in order for the outputs of deep neural networks not explode, the singular values of the mapping of each layer are not supposed to be deviating too much from one. Indeed, it is because it is extremely challenging to keep $\prod_{i=1}^{\text{num of layers}} \lambda_i$ from exploding or vanishing without keeping all of the $\lambda_i$ having unit norm. However, by design ResNet with shortcuts already have an identity mapping every few layers, which forces the mappings inside the shortcuts to have small operator norms. Hence, analyzing the network at zero initial point gives a decent characterization of the searching environment of the optimization algorithm.

Our results also shows that *the form of Hessian is more important than the existance of nonlinearities* when training the networks. The behaviors of the networks we studied are consistent across both linear and nonlinear structures, where networks with clearer Hessians are much easier to achieve lower training errors.

FIX

On the other hand, our experiments reveal that *orthogonal initialization (Saxe et al., 2013) is suboptimal*. Although better than Xavier initialization (Glorot & Bengio, 2010), the initial condition numbers of the networks still explode as the networks become deeper, which means the networks are still initialized on "bad" submanifolds that are hard to optimize using gradient descent.

## 3 MODEL

### 3.1 DEEP LINEAR NETWORKS

Deep linear networks are feed-forward neural networks that only contain linear units, which means their input-output mappings are simply linear transformations. Apparently, increasing their depths will not affect the representational power of the networks. However, linear networks with depth deeper than one show nonlinear dynamics of training (Saxe et al., 2013). As a result, analyzing the training of deep linear networks gives us a better understanding of the training of non-linear networks.

Much theoretical work has been done on deep linear networks. Kawaguchi (2016) extended the work of Choromanska et al. (2015a;b) and proved that with few assumptions, every local minimum point in deep linear networks is a global minimum point. This means that the difficulties in the training of deep linear networks mostly come from saddle points on the loss surfaces, which are also the main causes of slow learning in nonlinear networks (Pascanu et al., 2014).

Saxe et al. (2013) studied the dynamics of training using gradient descent. They found that for a special class of initial conditions, which could be obtained from greedy layerwise pre-training, the training time for a deep linear network with an infinity depth can still be finite. Furthermore, they found that by setting the initial weights to random orthogonal matrices (produced by performing QR or SVD decompositions on random Gaussian matrices), the network will still have a depth independent learning time. They argue that it is caused by the eigenvalue and singular value spectra of the end-to-end linear transformation. When using orthogonal initialization, the overall transformation is an orthogonal matrix, which has all the singular values equal to 1. In the meantime, when using scaled Gaussian initialization, most of the singular values are close to zero, making the network unsuitable for backpropagating errors. However, this explanation is not sufficient to prove that the training difficulty of orthogonal initialized networks is depth-invariant. It only gives us an intuition on why orthogonal initialization performs better than scaled Gaussian initialization.

Thus, we use deep linear networks to study the effect of shortcut connections. After adding the shortcuts, the overall model is still linear and the global minimum does not change.

### 3.2 NETWORK STRUCTURE

We first generalize a linear network by adding shortcuts to it to make it a *linear residual network*. We organize the network into $R$ *residual units*. The $r$-th residual unit consists of $L_r$ layers whose

weights are $W^{r,1}, \dots, W^{r,L_r-1}$, denoted as the *transformation path*, as well as a shortcut $S^r$ connecting from the first layer to the last one, denoted as the *shortcut path*. The input-output mapping can be written as

$$y = \prod_{r=1}^{R} (\prod_{l=1}^{L_r-1} W^{r,l} + S^r)x = Wx, \tag{4}$$

where $x \in \mathbb{R}^{d_x}, y \in \mathbb{R}^{d_y}, W \in \mathbb{R}^{d_y \times d_x}$. Here if $b \geq a$, $\prod_{i=a}^{b} W^i$ denotes $W^b W^{(b-1)} \cdots W^{(a+1)} W^a$, otherwise it denotes an identity mapping. The matrix $W$ represents the combination of all the linear transformations in the network. Note that by setting all the shortcuts to zeros, the network will go back to a $(\sum_r (L_r - 1) + 1)$-layer plain linear network.

Instead of analyzing the general form, we concentrate on a special kind of linear residual networks, where all the residual units are the same.

**Definition 2.** A linear residual network is called an *n-shortcut linear network* if

1. its layers have the same dimension (so that $d_x = d_y$);

2. its shortcuts are identity matrices;

3. its shortcuts have the same depth $n$.

The input-output mapping for such a network becomes

$$y = \prod_{r=1}^{R} (\prod_{l=1}^{n} W^{r,l} + I_{d_x})x = Wx, \tag{5}$$

where $W^{r,l} \in \mathbb{R}^{d_x \times d_x}$.

Then we add some activation functions to the networks. We concentrate on the case where activation functions are on the transformation paths, which is also the case in the latest ResNet (He et al., 2016).

**Definition 3.** An $n$-shortcut linear network becomes an *n-shortcut network* if element-wise activation functions $\sigma_{\text{pre}}(x), \sigma_{\text{mid}}(x), \sigma_{\text{post}}(x)$ are added at the transformation paths, where on a transformation path, $\sigma_{\text{pre}}(x)$ is added before the first weight matrix, $\sigma_{\text{mid}}(x)$ is added between two weight matrixes and $\sigma_{\text{post}}(x)$ is added after the last weight matrix.

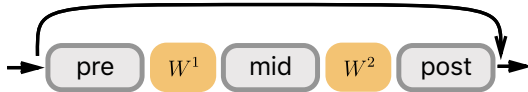

Figure 1: An example of different position for nonlinearities in a residual unit of a 2-shortcut network.

Note that $n$-shortcut linear networks are special cases of $n$-shortcut networks, where all the activation functions are identity mappings.

## 3.3 OPTIMIZATION

We denote the collection of all the variable weight parameters in an $n$-shortcut linear network as $\mathbf{w}$. Consider $m$ training samples $\{x^\mu, y^\mu\}, \mu = 1, \dots, m$. Using Frobenius loss, for an $n$-shortcut linear network, we define the loss function as follows:

$$L(\mathbf{w}) = \frac{1}{2m} \sum_{\mu=1}^{m} \|y^\mu - Wx^\mu\|_2^2 = \frac{1}{2m} \|Y - WX\|_F^2, \tag{6}$$

where $x^\mu, y^\mu$ are the $\mu$-th columns of $X, Y$, and $\|\cdot\|_F$ denotes the Frobenius norm. Using gradient descent with learning rate $\alpha$, we have the weights updating rules as

$$\Delta W^{r,l} = \alpha (W_{\text{after}}^r W_{\text{after}}^{r,l})^T (\Sigma^{YX} - W\Sigma^{XX})(W_{\text{before}}^{r,l} W_{\text{before}}^r)^T, \tag{7}$$

where $\Sigma^{XX}$ and $\Sigma^{YX}$ denote the input-input and the output-input correlation matrices, defined as

$$\Sigma^{XX} = \frac{1}{m}\sum_{\mu=1}^m x^\mu (x^\mu)^T \tag{8}$$

$$\Sigma^{YX} = \frac{1}{m}\sum_{\mu=1}^m y^\mu (x^\mu)^T. \tag{9}$$

Here $W_{\text{before}}^r, W_{\text{after}}^r$ denote the linear mappings before and after the $r$-th residual unit, $W_{\text{before}}^{r,l}, W_{\text{after}}^{r,l}$ denote the linear mappings before and after $W^{r,l}$ within the transformation path of the $r$-th residual unit. In other words, the overall transformation can be represented as

$$y = W_{\text{after}}^r (W_{\text{after}}^{r,l} W^{r,l} W_{\text{before}}^{r,l} + I_{d_x}) W_{\text{before}}^r x. \tag{10}$$

## 4 THEORETICAL STUDY

### 4.1 INITIAL POINT PROPERTIES

Before we analyze the initial point properties of $n$-shortcut networks, we have to choose the way to initialize them. ResNet uses MSRA initialization (He et al., 2015b). It is a kind of scaled Gaussian initialization that tries to keep the variances of signals along a transformation path, which is also the idea behind Xavier initialization (Glorot & Bengio, 2010). However, because of the shortcut paths, the output variance of the entire network will actually explode as the network becomes deeper. Batch normalization units partly solved this problem in ResNet, but still they cannot prevent the large output variance in a deep network.    NEW

A simple idea is to zero initialize all the weights, so that the output variances of residual units stay the same along the network. It is worth noting that as found in He et al. (2015a), the deeper ResNet has smaller magnitudes of layer responses. This phenomenon has been confirmed in our experiments. As illustrated in Figure 2 and Figure 3, the deeper a residual network is, the small its average Frobenius norm of weight matrixes is, both during the traning process and when the training ends. Also, Hardt & Ma (2016) proves that if all the weight matrixes have small norms, a linear residual network will have no critical points other than the global optimum.    NEW

All these evidences indicate that zero is spacial in a residual network: as the network becomes deeper, the training tends to end up around it. Thus, we are looking into the Hessian at zero. As the zero is a saddle point, in our experiments we use zero initialization with small random perturbations to escape from it. We first Xavier initialize the weight matrixes, and then multiply a small constant (0,01) to them.    NEW

We begin with the definition of $k$-th order stationary point.

**Definition 4.** Suppose function $f(x)$ admits $k$-th order Taylor expansion at point $x_0$. We say that the point $x_0$ is a $k$-th order stationary point of $f(x)$ if the corresponding $k$-th order Taylor expansion of $f(x)$ at $x = x_0$ is a constant: $f(x) = f(x_0) + o(\|x - x_0\|_2^k)$.

Now we state our main theorem, whose proof can be found in Appendix A.

**Theorem 1.** *Assume that $\sigma_{\text{mid}}(0) = \sigma_{\text{post}}(0) = 0$ and all of $\sigma_{\text{pre}}^{(k)}(0), \sigma_{\text{mid}}^{(k)}(0), \sigma_{\text{post}}^{(k)}(0), 1 \le k \le \max(n-1, 2)$ exist. For the loss function of an $n$-shortcut network, at point zero,*

1. *if $n \ge 2$, it is an $(n-1)$th-order stationary point. In particular, if $n \ge 3$, the Hessian is a zero matrix;*

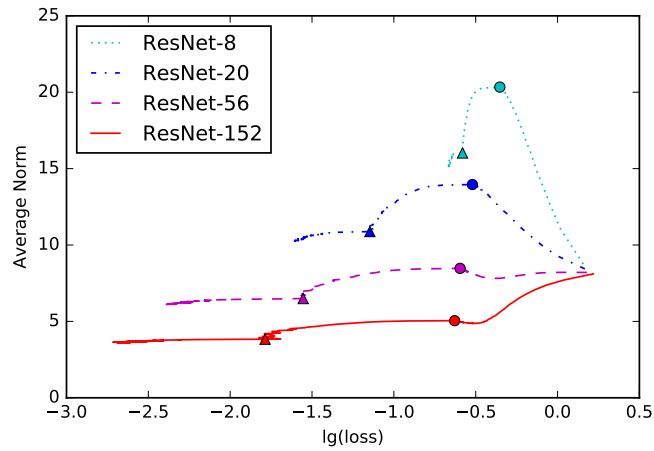

NEW

Figure 2: The average Frobenius norms of ResNets of different depths during the training process. The pre-ResNet implementation in `https://github.com/facebook/fb.resnet.torch` is used. The learning rate is initialized to 0.1, decreased to 0.01 at the 81st epoch (marked with circles) and decreased to 0.001 at the 122nd epoch (marked with triangles). Each model is trained for 200 epochs.

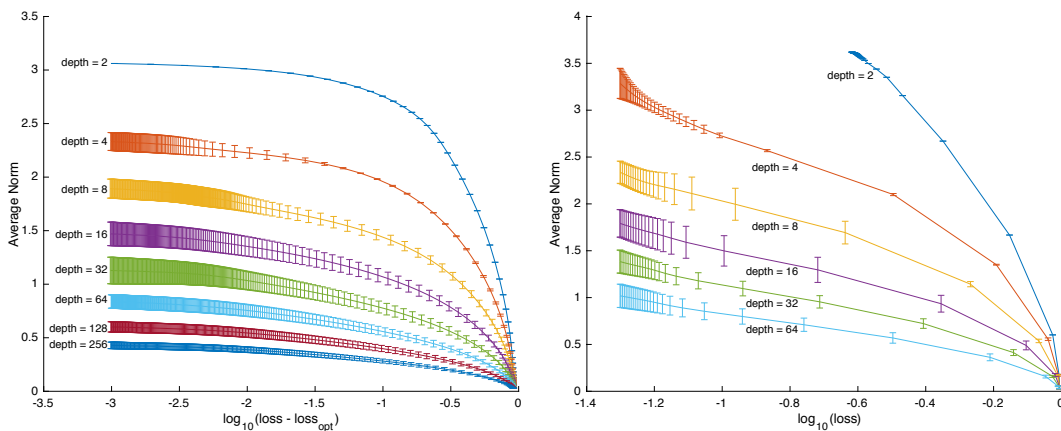

NEW

Figure 3: The average Frobenius norms of 2-shortcut networks of different depths during the training process when zero initialized. **Left**: Without nonlinearities. **Right**: With ReLUs at mid positions.

2. *if $n = 2$, the Hessian can be written as*

$$H = \begin{bmatrix} \mathbf{0} & A^T & & & \\ A & \mathbf{0} & & & \\ & & \mathbf{0} & A^T & \\ & & A & \mathbf{0} & \\ & & & & \ddots \end{bmatrix}, \tag{11}$$

*whose condition number is*

$$\mathrm{cond}(H) = \sqrt{\mathrm{cond}((\Sigma^X \sigma_{\mathrm{pre}}(X) - \Sigma^Y \sigma_{\mathrm{pre}}(X))^T (\Sigma^X \sigma_{\mathrm{pre}}(X) - \Sigma^Y \sigma_{\mathrm{pre}}(X)))}, \tag{12}$$

*where $A$ only depends on the training set and the activation functions. Except for degenerate cases, it is a* strict *saddle point (Ge et al., 2015).*

3. *if $n = 1$, the Hessian can be written as*

$$H = \begin{bmatrix} B & A^T & A^T & \cdots & A^T \\ A & B & A^T & \cdots & A^T \\ A & A & B & & \vdots \\ \vdots & \vdots & & \ddots & A^T \\ A & A & \cdots & A & B \end{bmatrix} \tag{13}$$

*where $A, B$ only depend on the training set and the activation functions.*

Theorem 1 shows that the condition numbers of 2-shortcut networks are depth-invariant with a nice structure of eigenvalues. Indeed, the eigenvalues of the Hessian $H$ at the zero initial point are multiple copies of $\pm \sqrt{\mathrm{eigs}(A^T A)}$, and the number of copies is equal to the number of shortcut connections.

The Hessian at zero initial point for the 1-shortcut linear network follows block Toeplitz structure, which has been well studied in the literature. In particular, its condition number tends to explode as the number of layers increase (Gray, 2006).

The assumptions hold for most activation functions including tanh, symmetric sigmoid and ReLU (Nair & Hinton, 2010). Note that although ReLU does not have derivatives at zero, one may do a local polynomial approximation to yield $\sigma^{(k)}, 1 \le k \le \max(n-1, 2)$.

To get intuitive explanations of the theorems, imagine changing parameters in an $n$-shortcut network. One has to change at least $n$ parameters to make any difference in the loss. So zero is an $(n-1)$th-order stationary point. Notice that the higher the order of a stationary point, the more difficult for a first order method to escape from it.

On the other hand, if $n = 2$, one will have to change two parameters in the same residual unit but different weight matrices to affect the loss, leading to a clear block diagonal Hessian.

## 4.2 LEARNING DYNAMICS

To understand Equation (7) better, we can take $n$-shortcut linear networks to two extremes. First, when $n = 1$, let $V^{r,1} = W^{r,1} + I_{d_x}, r = 1, \ldots, R-1$. As $I_{d_x}$ is a constant, we have

$$\Delta V^{r,1} = \alpha (\prod_{r'=r+1}^{R-1} V^{r',1})^T (\Sigma^{YX} - (\prod_{r'=1}^{R-1} V^{r',1}) \Sigma^{XX}) (\prod_{r'=1}^{r-1} V^{r',1})^T, \tag{14}$$

which can be seen as a linear network with identity initialization, a special case of orthogonal initialization, if the original 1-shortcut network is zero initialized.

On the other side, if the number of shortcut connections $R = 1$, the shortcut will only change the distribution of the output training set from $Y$ to $Y - X$. These two extremes are illustrated in Figure 4

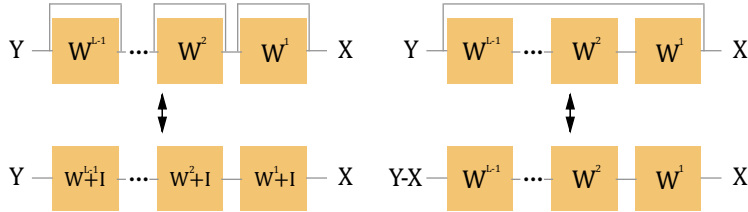

Figure 4: Equivalents of two extremes of $n$-shortcut linear networks. 1-shortcut linear networks are equivalent to linear networks with identity initialization, while skip-all shortcuts will only change the effective dataset outputs.

### 4.3 LEARNING RESULTS

The optimal weights of an $n$-shortcut linear network can be easily computed via least squares, which leads to

$$W = YX^T(XX^T)^{-1} = \Sigma^{YX}(\Sigma^{XX})^{-1}, \tag{15}$$

and the minimum of the loss function is

$$L_{\min} = \frac{1}{2m}\|Y - \Sigma^{YX}(\Sigma^{XX})^{-1}X\|_F^2, \tag{16}$$

where $\|\cdot\|_F$ denotes the Frobenius norm and $(\Sigma^{XX})^{-1}$ denotes any kind of generalized inverse of $\Sigma^{XX}$. So given a training set, we can pre-compute its $L_{\min}$ and use it to evaluate any $n$-shortcut linear network.

## 5 EXPERIMENTS

We compare networks with Xavier initialization (Glorot & Bengio, 2010), networks with orthogonal initialization (Saxe et al., 2013) and 2-shortcut networks with zero initialization. The training dynamics of 1-shortcut networks are similar to that of linear networks with orthogonal initialization in our experiments. Setup details can be found in Appendix B.

### 5.1 INITIAL POINT

We first compute the initial condition numbers for different kinds of linear networks with different depths.

As can be seen in Figure 5, **2-shortcut linear networks have constant condition numbers as expected.** On the other hand, when using Xavier or orthogonal initialization in linear networks, the initial condition numbers will go to infinity as the depths become infinity, making the networks hard to train. This also explains why orthogonal initialization is helpful for a linear network, as its initial condition number grows slower than the Xavier initialization.

### 5.2 LEARNING DYNAMICS

Having a good beginning does not guarantee an easy trip on the loss surface. In order to depict the loss surfaces encountered from different initial points, we plot the maxima and $10^{\text{th}}$ percentiles (instead of minima, as they are very unstable) of the absolute values of Hessians eigenvalues at different losses.

As shown in Figure 6 and Figure 7, the condition numbers of 2-shortcut networks at different losses are always smaller, especially when the loss is large. Also, notice that the condition numbers roughly evolved to the same value for both orthogonal and 2-shortcut linear networks. This may be explained by the fact that the minimizers, as well as any point near them, have similar condition numbers.

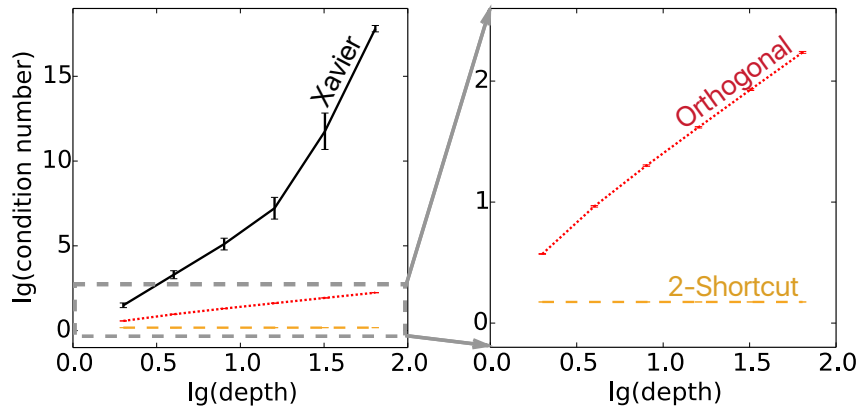

Figure 5: Initial condition numbers of Hessians for different linear networks as the depths of the networks increase. Means and standard deviations are estimated based on 10 runs.

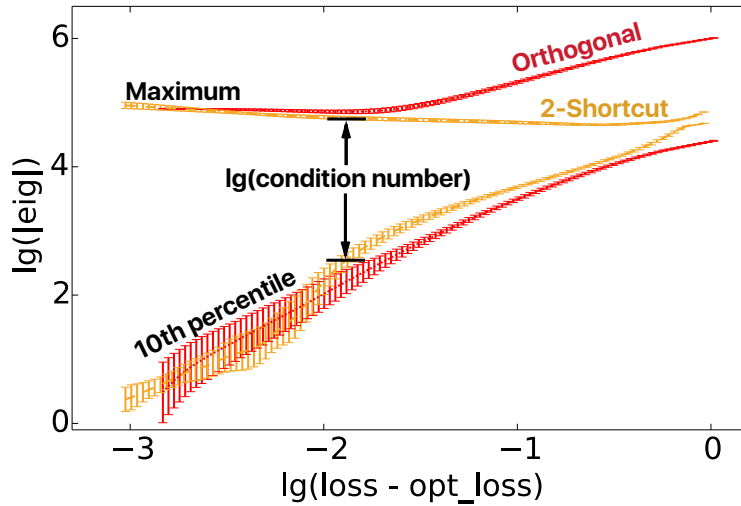

Figure 6: Maxima and 10[th] percentiles of absolute values of eigenvalues at different losses when the depth is 16. For each run, eigenvalues at different losses are calculated using linear interpolation.

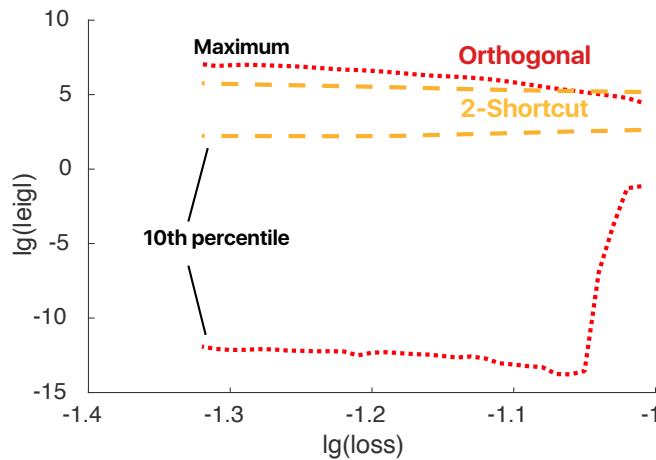

NEW

Figure 7: Maxima and 10[th] percentiles of absolute values of eigenvalues at different losses when the depth is 16. Eigenvalues at different losses are calculated using linear interpolation.

Another observation is the changes of negative eigenvalues ratios. *Index* (ratio of negative eigenvalues) is an important characteristic of a critical point. Usually for the critical points of a neural network, the larger the loss the larger the index (Dauphin et al., 2014). In our experiments, the index of a 2-shortcut network is always smaller, and drops dramatically at the beginning, as shown in Figure 8, left. This might make the networks tend to stop at low critical points.

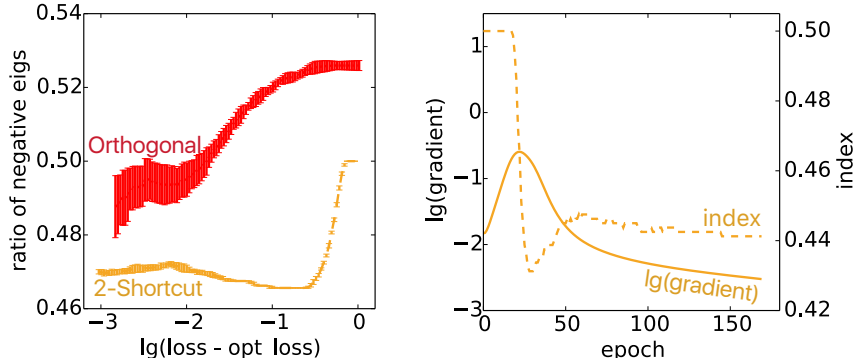

Figure 8: **Left**: ratio of negative eigenvalues at different losses when the depth is 16. For each run, indexes at different losses are calculated using linear interpolation. **Right**: the dynamics of gradient and index of a 2-shortcut linear network in a single run. The gradient reaches its maximum while the index drops dramatically, indicating moving toward negative curvature directions.

This is because the initial point is near a saddle point, thus it tends to go towards negative curvature directions, eliminating some negative eigenvalues at the beginning. This phenomenon matches the observation that the gradient reaches its maximum when the index drops dramatically, as shown in Figure 8, right.

## 5.3 LEARNING RESULTS

We run different networks for 1000 epochs using different learning rates at log scale, and compare the average final losses of the optimal learning rates.

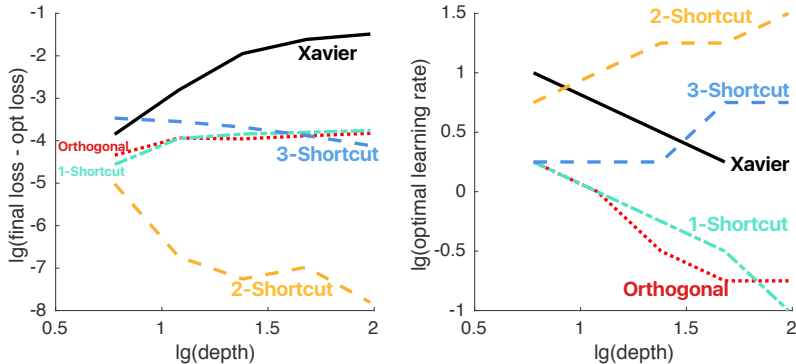

NEW

Figure 9: **Left**: Optimal Final losses of different linear networks. **Right**: Corresponding optimal learning rates. When the depth is 96, the final losses of Xavier with different learning rates are basically the same, so the optimal learning rate is omitted as it is very unstable.

Figure 9 shows the results for linear networks. Just like their depth-invariant initial condition numbers, the final losses of 2-shortcut linear networks stay close to optimal as the networks become deeper. Higher learning rates can also be applied, resulting in fast learning in deep networks.

Then we add ReLUs to the *mid* positions of the networks. To make a fair comparison, the numbers of ReLU units in different networks are the same when the depths are the same, so 1-shortcut and 3-shortcut networks are omitted. The result is shown in Figure 10.

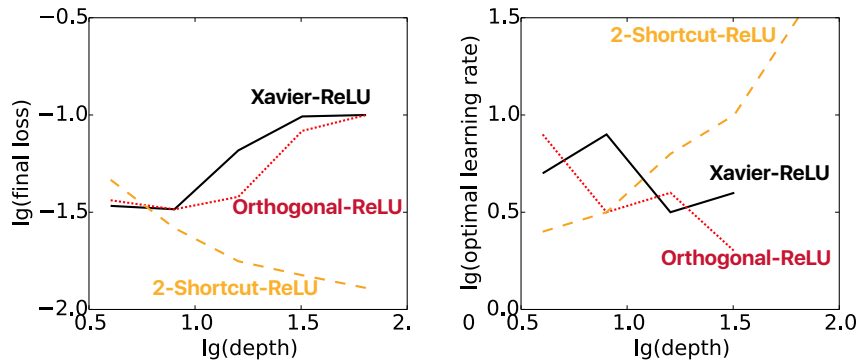

Figure 10: **Left**: Optimal Final losses of different networks with ReLUs in *mid* positions. **Right**: Corresponding optimal learning rates. Note that as it is hard to compute the minimum losses with ReLUs, we plot the $\log_{10}$(final loss) instead of $\log_{10}$(final loss − optimal loss). When the depth is 64, the final losses of Xavier-ReLU and orthogonal-ReLU with different learning rates are basically the same, so the optimal learning rates are omitted as they are very unstable.

Note that because of the nonlinearities, the optimal losses vary for different networks with different depths. It is usually thought that deeper networks can represent more complex models, leading to smaller optimal losses. However, our experiments show that linear networks with Xavier or orthogonal initialization have difficulties finding these optimal points, while 2-shortcut networks find these optimal points easily as they did without nonlinear units.

## 6 FUTURE DIRECTIONS

Further studies should concentrate on the behavior of shortcut connections on convolution networks, as well as the influences of batch normalization units (Ioffe & Szegedy, 2015) in ResNet. Meanwhile, it would be very interesting to extend the insights obtained in this paper to recurrent neural networks such as LSTM (Sainath et al., 2013).

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

## A PROOFS OF THEOREMS

**Definition 5.** The elements in Hessian of an $n$-shortcut network is defined as

$$H_{\text{ind}(w_1),\text{ind}(w_2)} = \frac{\partial^2 L}{\partial w_1 \partial w_2}, \tag{17}$$

where $L$ is the loss function, and the indices $\text{ind}(\cdot)$ is ordered lexicographically following the four indices $(r, l, j, i)$ of the weight variable $w_{i,j}^{r,l}$. In other words, the priority decreases along the index of shortcuts, index of weight matrix inside shortcuts, index of column, and index of row.

Note that the collection of all the weight variables in the $n$-shortcut network is denoted as $\mathbf{w}$. We study the behavior of the loss function in the vicinity of $\mathbf{w} = \mathbf{0}$.

**Lemma 1.** *Assume that* $w_1 = w_{i_1,j_1}^{r_1,l_1}, \cdots, w_N = w_{i_N,j_N}^{r_N,l_N}$ *are $N$ parameters of an $n$-shortcut network. If* $\frac{\partial^2 L}{\partial w_1 \cdots \partial w_N}\Big|_{\mathbf{w}=\mathbf{0}}$ *is nonzero, there exists $r$ and $k_1, \cdots, k_n$ such that $r_{k_m} = r$ and $l_{k_m} = m$ for $m = 1, \cdots, n$.*

*Proof.* Assume there does not exist such $r$ and $k_1, \cdots, k_n$, then for all the shortcut units $r = 1, \cdots, R$, there exists a weight matrix $l$ such that none of $w_1, \cdots, w_N$ is in $W^{r,l}$, so all the transformation paths are zero, which means $W = I_{d_x}$. Then $\frac{\partial^2 L}{\partial w_1 \cdots \partial w_N}\Big|_{\mathbf{w}=\mathbf{0}} = 0$, leading to a contradiction. $\qquad\square$

**Lemma 2.** *Assume that* $w_1 = w_{i_1,j_1}^{r_1,l_1}, w_2 = w_{i_2,j_2}^{r_2,l_2}, r_1 \leq r_2$. *Let $L_0(w_1, w_2)$ denotes the loss function with all the parameters except $w_1$ and $w_2$ set to 0, $w_1' = w_{i_1,j_1}^{1,l_1}, w_2' = w_{i_2,j_2}^{1+\mathbb{1}(r_1 \neq r_2),l_2}$. Then $\frac{\partial^2 L_0(w_1,w_2)}{\partial w_1 \partial w_2}\Big|_{(w_1,w_2)=\mathbf{0}} = \frac{\partial^2 L_0(w_1',w_2')}{\partial w_1' \partial w_2'}\Big|_{(w_1',w_2')=\mathbf{0}}$.*

*Proof.* As all the residual units expect unit $r_1$ and $r_2$ are identity transformations, reordering residual units while preserving the order of units $r_1$ and $r_2$ will not affect the overall transformation, i.e. $L_0(w_1, w_2)|_{w_1=a,w_2=b} = L_0'(w_1', w_2')|_{w_1'=a,w_2'=b}$. So $\frac{\partial^2 L_0(w_1,w_2)}{\partial w_1 \partial w_2}\Big|_{(w_1,w_2)=\mathbf{0}} = \frac{\partial^2 L_0(w_1',w_2')}{\partial w_1' \partial w_2'}\Big|_{(w_1',w_2')=\mathbf{0}}$. $\qquad\square$

*Proof of Theorem 1.* Now we can prove Theorem 1 with the help of the previously established lemmas.

1. Using Lemma 1, for an $n$-shortcut network, at zero, all the $k$-th order partial derivatives of the loss function are zero, where $k$ ranges from 1 to $n-1$. Hence, the initial point zero is a $(n-1)$th-order stationary point of the loss function.

2. Consider the Hessian in $n = 2$ case. Using Lemma 1 and Lemma 2, the form of Hessian can be directly written as Equation (11), as illustrated in Figure 11.

   So we have

   $$\text{eigs}(H) = \text{eigs}(\begin{bmatrix} \mathbf{0} & A^T \\ A & \mathbf{0} \end{bmatrix}) = \pm\sqrt{\text{eigs}(A^T A)}. \tag{18}$$

   Thus $\text{cond}(H) = \sqrt{\text{cond}(A^T A)}$, which is depth-invariant. Note that the dimension of $A$ is $d_x^2 \times d_x^2$.

   To get the expression of $A$, consider two parameters that are in the same residual unit but different weight matrices, i.e. $w_1 = w_{i_1,j_1}^{r,2}, w_2 = w_{i_2,j_2}^{r,1}$.

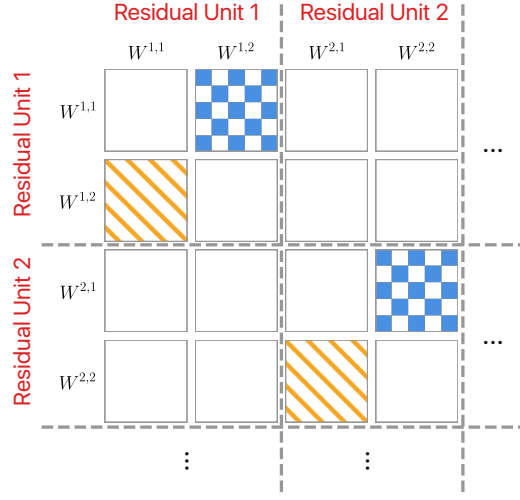

Figure 11: The Hessian in $n = 2$ case. It follows from Lemma 1 that only off-diagonal subblocks in each diagonal block, i.e., the blocks marked in orange (slash) and blue (chessboard), are non-zero. From Lemma 2, we conclude the translation invariance and that all blocks marked in orange (slash) (resp. blue (chessboard)) are the same. Given that the Hessian is symmetric, the blocks marked in blue and orange are transposes of each other, and thus it can be directly written as Equation (11).

If $j_1 = i_2$, we have

$$
\begin{aligned}
A_{(j_1-1)d_x+i_1,(j_2-1)d_x+i_2} &= \left.\frac{\partial^2 L}{\partial w_1 \partial w_2}\right|_{\mathbf{w}=\mathbf{0}} \\
&= \left.\frac{\partial^2 \sum_{\mu=1}^m \frac{1}{2m}(y_{i_1}^\mu - x_{i_1}^\mu - \sigma_{\text{post}}(w_1\sigma_{\text{mid}}(w_2\sigma_{\text{pre}}(x_{j_2}^\mu))))^2}{\partial w_1 \partial w_2}\right|_{\mathbf{w}=\mathbf{0}} \\
&= \frac{\sigma'_{\text{mid}}(0)\sigma'_{\text{post}}(0)}{m}\sum_{\mu=1}^m \sigma_{\text{pre}}(x_{j_2}^\mu)(x_{i_1}^\mu - y_{i_1}^\mu).
\end{aligned}
\tag{19}
$$

Else, we have $A_{(j_1-1)d_x+i_1,(j_2-1)d_x+i_2} = 0$.

Noting that $A_{(j_1-1)d_x+i_1,(j_2-1)d_x+i_2}$ in fact only depends on the two indices $i_1, j_2$ (with a small difference depending on whether $j_1 = i_2$), we make a $d_x \times d_x$ matrix with rows indexed by $i_1$ and columns indexed by $j_2$, and the entry at $(i_1, j_2)$ equal to $A_{(j_1-1)d_x+i_1,(j_2-1)d_x+i_2}$. Apparently, this matrix is equal to $\sigma'_{\text{mid}}(0)\sigma'_{\text{post}}(0)(\Sigma^{X\sigma_{\text{pre}}(X)} - \Sigma^{Y\sigma_{\text{pre}}(X)})$ when $j_1 = i_2$, and equal to the zero matrix when $j_1 \neq i_2$.

To simplify the expression of $A$, we rearrange the columns of $A$ by a permutation matrix, i.e.

$$A' = AP, \tag{20}$$

where $P_{ij} = 1$ if and only if $i = ((j-1) \bmod d_x)d_x + \lceil \frac{j}{d_x} \rceil$. Basically it permutes the $i$-th column of $A$ to the $j$-th column.

Then we have

$$
A = \sigma'_{\text{mid}}(0)\sigma'_{\text{post}}(0)\begin{bmatrix} \Sigma^{X\sigma_{\text{pre}}(X)} - \Sigma^{Y\sigma_{\text{pre}}(X)} & & \\ & \ddots & \\ & & \Sigma^{X\sigma_{\text{pre}}(X)} - \Sigma^{Y\sigma_{\text{pre}}(X)} \end{bmatrix} P^T.
\tag{21}
$$

So the eigenvalues of $H$ becomes

$$\text{eigs}(H) = \pm\sigma'_{\text{mid}}(0)\sigma'_{\text{post}}(0)\sqrt{\text{eigs}((\Sigma^X\sigma_{\text{pre}}(X) - \Sigma^Y\sigma_{\text{pre}}(X))^T(\Sigma^X\sigma_{\text{pre}}(X) - \Sigma^Y\sigma_{\text{pre}}(X)))}, \tag{22}$$

which leads to Equation (12).

3. Now consider the Hessian in the $n = 1$ case. Using Lemma 2, the form of Hessian can be directly written as Equation (13).

To get the expressions of $A$ and $B$ in $\sigma_{\text{pre}}(x) = \sigma_{\text{post}}(x) = x$ case, consider two parameters that are in the same residual units, i.e. $w_1 = w_{i_1,j_1}^{r,1}, w_2 = w_{i_2,j_2}^{r,1}$.

We have

$$B_{(j_1-1)d_x+i_1,(j_2-1)d_x+i_2} = \frac{\partial^2 L}{\partial w_1 \partial w_2}\bigg|_{\mathbf{w}=\mathbf{0}} \tag{23}$$

$$= \begin{cases} \frac{1}{m}\sum_{\mu=1}^m x_{j_1}^\mu x_{j_2}^\mu & i_1 = i_2 \\ 0 & i_1 \neq i_2 \end{cases} \tag{24}$$

Rearrange the order of variables using $P$, we have

$$B = P \begin{bmatrix} \Sigma^{XX} & & \\ & \ddots & \\ & & \Sigma^{XX} \end{bmatrix} P^T. \tag{25}$$

Then consider two parameters that are in different residual units, i.e. $w_1 = w_{i_1,j_1}^{r_1,1}, w_2 = w_{i_2,j_2}^{r_2,1}, r_1 > r_2$.

We have

$$A_{(j_1-1)d_x+i_1,(j_2-1)d_x+i_2} = \frac{\partial^2 L}{\partial w_1 \partial w_2}\bigg|_{\mathbf{w}=\mathbf{0}} \tag{26}$$

$$= \begin{cases} \frac{1}{m}\sum_{\mu=1}^m (x_{i_1}^\mu - y_{i_1}^\mu)x_{j_2}^\mu + x_{j_1}^\mu x_{j_2}^\mu & j_1 = i_2, i_1 = i_2 \\ \frac{1}{m}\sum_{\mu=1}^m (x_{i_1}^\mu - y_{i_1}^\mu)x_{j_2}^\mu & j_1 = i_2, i_1 \neq i_2 \\ \frac{1}{m}\sum_{\mu=1}^m x_{j_1}^\mu x_{j_2}^\mu & j_1 \neq i_2, i_1 = i_2 \\ 0 & j_1 \neq i_2, i_1 \neq i_2 \end{cases} \tag{27}$$

In the same way, we can rewrite $A$ as

$$A = \begin{bmatrix} \Sigma^{XX} - \Sigma^{YX} & & \\ & \ddots & \\ & & \Sigma^{XX} - \Sigma^{YX} \end{bmatrix} P^T + B. \tag{28}$$

$\square$

# B   EXPERIMENT SETUP

We took the experiments on whitened versions of MNIST. 10 greatest principal components are kept for the dataset inputs. The dataset outputs are represented using one-hot encoding. The network was trained using gradient descent. For every epoch, the Hessians of the networks were calculated using the method proposed in (Bishop, 1992). As the $|\lambda|_{\text{min}}$ of Hessian is usually very unstable, we calculated $\frac{|\lambda|_{\text{max}}}{|\lambda|_{(0.1)}}$ to represent condition number instead, where $|\lambda|_{(0.1)}$ is the $10^{\text{th}}$ percentile of the absolute values of eigenvalues.

As *pre*, *mid* or *post* positions are not defined in linear networks without shortcuts, when comparing Xavier or orthogonal initialized linear networks to 2-shortcut networks, we added ReLUs at the same positions in linear networks as in 2-shortcuts networks.

