# Peer review of "Demystifying ResNet"

_ICLR 2017 — rejected_

[Public Comment · Ashok Vardhan Makkuva · 22 Nov 2016]
**Definition of condition number**

Hi Authors,

I went through your paper in detail and it was very good linking the Hessian at the zero initial point to the success of the ResNet. However, I am a little unsure of the definition of the condition number in the paper. Across several standard refernces, the condition number is defined as the ratio the maximum and minimum singular values where as you defined it as $ |\lambda|_max / |\lambda|_min  $.  Unless the matrix, which is the Hessian H in your case, is positive semi-definite, these two are not the same and the spectra of the singular values and the eigen values can be quite different in general. Especially, for n=2, the initial point zero is a saddle point as stated in your theorem and as such the Hessian is not positive-definite. Please let me know if I am missing out on something and whether or not this modification affects the analysis in the paper. Thanks!

[Official Review · AnonReviewer1 · rating 4 · confidence 5 · 15 Dec 2016]
**This paper tries to explain the reason behind the success of ResNet architecture through theoretical arguments and experimental results. Unfortunately, theoretical arguments and experimental results are not enough to support the main claims of the paper.**

ResNet and other architectures that use shortcuts have shown empirical success in several domains and therefore, studying the optimization for such architectures is very valuable. This paper is an attempt to address some of the properties of networks that use shortcuts. Some of the experiments in the paper are interesting. However, there are two main issues with the current paper:

1- linear vs non-linear: I think studying linear networks is valuable but we should be careful not to extend the results to networks with non-linear activations without enough evidence. This is especially true for Hessian as the Hessian of non-linear networks have very large condition number (see the ICLR submission "Singularity of Hessian in Deep Learning") even in cases where the optimization is not challenging. Therefore, I don't agree with the claims in the paper on non-linear networks. Moreover, one plot on MNIST is not enough to claim that non-linear networks behave similar to linear networks.

2- Hessian at zero initial point: The explanation of why we should be interested in Hessain at zero initial point is not acceptable. The zero initial point is not interesting because it is a very particular point that cannot tell us about the Hessian during optimization.

[Official Review · AnonReviewer3 · rating 5 · confidence 3 · 16 Dec 2016]
**this paper has nice observation**

This paper studies the optimization issue of linear ResNet, and shows mathematically that for 2-shortcuts and zero initialization, the Hessian has condition number independent of depth. I skimmed through the proof but have not checked them carefully. 

This result is a nice observation for training deep linear networks.  But I do not think the paper has fully resolved the linear vs nonlinear issue. Some question:

1. Though the revision has added some results using ReLU units, it seems it is only added to the mid positions of the network (sec 5.3), is this how it is typically done in ResNet? Moreover, ReLU is not differentiable at zero point, which does not satisfy the condition you had in Theorem 1. Why not use differentiable activations like sigmoid or tanh?

2. From equation (22) in the appendix, it seems for nonlinear activations, the condition number depends on the derivative \sigma^\prime at 0. Therefore, if we use tanh which has derivative 1 at zero, the condition number is the same for linear and tanh activations. But this probably is not enough to explain the bit difference in performance or optimization for linear and nonlinear networks, or how the situations evolve after learning the 0 point.

3. As for the success of ResNet (or convnets in general) in computer vision, I believe there are more types of nonlinearity such as pooling? Can the result here generalizes to pooling as well?

Minor: 
- sec 1 last paragraph, low approximation error typically means more powerful model class and better training error, but not necessarily better test error
- sec 4.1 what do you mean by "zero initialization with small random perturbations"? why not exactly zero initialization, how large is the random perturbation?

[Official Review · AnonReviewer2 · rating 4 · confidence 4 · 18 Dec 2016]
**not ready yet**

I think the write-up can be improved. The results of the paper also might be somewhat misleading. The behavior for when weights are 0 is not revealing of how the model works in general. I think the work also underestimates the effect of the nonlinearities on the learning dynamics of the model.

[Final Decision · Program Chairs · 06 Feb 2017]
**ICLR committee final decision**

This paper endeavors to offer theoretical explanations of the performance of ResNet. Providing better theoretical understanding of existing empirically powerful architectures is very important work and I commend the authors for tackling this. Unfortunately, this paper falls short in its current form: the particular choices and restrictions made (0 weights, linear regime) limit applicability to ResNet, and do not seem to offer insights sufficient to capture the causes of ResNet's performance.